# Microwave Measurements of Electromagnetic Properties of Materials

**DOI:** 10.3390/ma14175097

**Published:** 2021-09-06

**Authors:** Jerzy Krupka

**Affiliations:** Institute of Microelectronics and Optoelectronics, Warsaw University of Technology, 00662 Warsaw, Poland; j.krupka@imio.pw.edu.pl; Tel.: +48-50-123-7390; Fax: +48-22-825-3055

**Keywords:** complex permittivity, initial permeability, permeability tensor, contactless resistivity measurement, dielectric resonator, superconductors, two fluid model

## Abstract

A review of measurement methods of the basic electromagnetic parameters of materials at microwave frequencies is presented. Materials under study include dielectrics, semiconductors, conductors, superconductors, and ferrites. Measurement methods of the complex permittivity, the complex permeability tensor, and the complex conductivity and related parameters, such as resistivity, the sheet resistance, and the ferromagnetic linewidth are considered. For dielectrics and ferrites, the knowledge of their complex permittivity and the complex permeability at microwave frequencies is of practical interest. Microwave measurements allow contactless measurements of their resistivity, conductivity, and sheet resistance. These days contactless conductivity measurements have become more and more important, due to the progress in materials technology and the development of new materials intended for the electronic industry such as graphene, GaN, and SiC. Some of these materials, such as GaN and SiC are not measurable with the four-point probe technique, even if they are conducting. Measurement fixtures that are described in this paper include sections of transmission lines, resonance cavities, and dielectric resonators.

## 1. Introduction

Knowledge of the electromagnetic properties of materials is important for their manufactures and users. Different measurement techniques are used which depend on the frequency of interest. Interaction of materials with electromagnetic radiation is described by Maxwell’s equations, where material properties are generally described by frequency-dependent permittivity, permeability, and conductivity tensors. Microwave measurement techniques operating at a frequency range from $10^{9}$ to $10^{11}$ Hz are important for two reasons. Firstly because many electronic devices and systems operate at the microwave frequency range e.g., satellite TV, telecommunication systems, radars, GPS. Secondly because some basic electric parameters e.g., resistivity, can be conveniently measured with contactless microwave techniques. The review of the microwave measurement methods presented in this paper is based on over 40 years of research work performed by the author.

## 2. Basic Definitions

At the frequency domain, the part of the complex permittivity related to the dielectric polarization mechanisms, is defined as a tensor quantity ${\overset{=}{\varepsilon }}_{d}$ describing the relationship: $D={\overset{=}{\varepsilon }}_{d}E$ between the electric displacement D and the electric field E vectors. Similarly, permeability tensor $\overset{=}{\mu }$ describes the relationship ***B***
$=\overset{=}{\mu }H$ between the magnetic induction ***B*** and the magnetic field ***H*** vectors. Finally, the complex conductivity tensor $\overset{=}{\sigma }$ describes the relationship between the electric current density j and the electric field E vectors $j=\overset{=}{\sigma }E$. For time-harmonic electromagnetic fields permittivity, permeability and conductivity are complex quantities and their imaginary parts describe phase shifts between appropriate components of the electromagnetic field vectors. In this paper the following notations are used ${\overset{=}{\varepsilon }}_{d}=Re\left({\overset{=}{\varepsilon }}_{d}\right)-j\;Im\left({\overset{=}{\varepsilon }}_{d}\right)$, $\overset{=}{\mu }=Re\left(\overset{=}{\mu }\right)-j\;Im\left(\overset{=}{\mu }\right)$, and $\overset{=}{\sigma }=Re\left(\overset{=}{\sigma }\right)-j\;Im\left(\overset{=}{\sigma }\right)$.

### 2.1. Permittivity and Conductivity Tensors

In Maxwell’s curl equation for time-harmonic e-m waves: $curl\;H=j+j\omega {\overset{=}{\varepsilon }}_{d}E=j\omega ({\overset{=}{\varepsilon }}_{d}-j\overset{=}{\sigma }/\omega$), one can combine dielectric permittivity and conductivity tensors into one permittivity tensor as follows.(1)$$\overset{=}{\varepsilon }={\overset{=}{\varepsilon }}_{d}-\frac{j}{\omega }\overset{=}{\sigma }=Re\left({\overset{=}{\varepsilon }}_{d}\right)-\;Im\left(\overset{=}{\sigma }\right)-j(Im\left({\overset{=}{\varepsilon }}_{d})+Re(\overset{=}{\sigma })/\omega \right)$$where ω—angular frequency.

In practice, the dielectric properties of a material are characterized by the dimensionless relative permittivity ${\overset{=}{\varepsilon }}_{r}$ which is defined as ${\overset{=}{\varepsilon }}_{r}=\overset{=}{\varepsilon }/\varepsilon_{0}$ where $\varepsilon_{0}$ denotes permittivity of vacuum $\varepsilon_{0}=8.854187\times 10^{-12}$ F/m.

The effective permittivity tensor elements for lossy materials are complex quantities, and their imaginary parts depend on both the dielectric and the conductor losses: $Im\left(\overset{=}{\varepsilon }\right)=Im\left({\overset{=}{\varepsilon }}_{d}\right)+Re\left(\overset{=}{\sigma }\right)/\omega$.

For linear, non-conducting dielectric medium permittivity tensor is symmetric, in most cases. In such a case, a specific coordinate system exists in which a relative permittivity tensor takes the diagonal form (2).(2)$${\overset{=}{\varepsilon }}_{r}=[\begin{array}{ccc} \varepsilon_{11} & 0 & 0\\ 0 & \varepsilon_{22} & 0\\ 0 & 0 & \varepsilon_{33} \end{array}]$$


If any two of the three permittivity tensor components are identical, medium is said to be uniaxially anisotropic, and if all three components are identical medium is dielectrically isotropic. The square root of the relative permittivity for the non-magnetic medium is named as the refractive index *n*. For an isotropic dielectric, all permittivity tensor components are identical and material can be characterized by one complex number called a scalar permittivity $\varepsilon_{r}$ or a dielectric constant. The ratio of the imaginary part and the real part of permittivity is called the dielectric loss tangent
$\tan\delta =Im(\varepsilon_{r})/Re(\varepsilon_{r})$.

In the absence of a static magnetic field, the conductivity tensor for an anisotropic conducting medium is symmetric, and, similarly to permittivity tensor, can be diagonalized. However, in the presence of a static magnetic field, the conductivity tensor can be represented as a sum of symmetric and antisymmetric parts. Conductivity and resistivity of a sample subjected to a static magnetic field aligned with the z-axis of a Cartesian coordinate system, which is associated with the symmetric part of conductivity tensor, take the following form.
(3)$$\bar{\bar{\sigma }}=\left|\begin{array}{ccc} \sigma_{xx} & \sigma_{xy} & 0\\ \sigma_{yx} & \sigma_{yy} & 0\\ 0 & 0 & \sigma_{zz}^{} \end{array}\right|,\;\bar{\bar{\rho }}=\left|\begin{array}{ccc} \rho_{xx} & \rho_{xy} & 0\\ \rho_{yx} & \rho_{yy} & 0\\ 0 & 0 & \rho_{zz}^{} \end{array}\right|$$

The resistivity tensor is the inverse of the conductivity tensor $\bar{\bar{\rho }}={\left(\bar{\bar{\sigma }}\right)}^{-1}$. $\sigma_{xx}=\sigma_{yy},\sigma_{yx}=-\sigma_{xy}$. The following relationships between conductivity and resistivity tensor components hold.
(4)$$\rho_{xx}=\rho_{yy}=\frac{\sigma_{xx}}{\sigma_{xx}^{2}+\sigma_{xy}^{2}},\;\rho_{yx}=-\rho_{xy}=\frac{\sigma_{xy}}{\sigma_{xx}^{2}+\sigma_{xy}^{2}},\;\rho_{zz}=\frac{1}{\sigma_{zz}}$$

### 2.2. Permeability Tensor

Non-reciprocal properties of ferrite devices, such as circulators or isolators are related to the ferromagnetic resonance (FMR) phenomenon [1]. Permeability of ferromagnetic material uniformly biased with a static magnetic field can be represented as a tensor quantity (5) [2].
(5)$$\overset{=}{\mu }=\mu_{0}\left[\begin{array}{ccc} \mu & j\kappa & 0\\ -j\kappa & \mu & 0\\ 0 & 0 & \mu_{\left|\right|} \end{array}\right]$$

The components of the relative permeability tensor ${\overset{=}{\mu }}_{r}=\overset{=}{\mu }/\mu_{o}$ take the following form [2]:(6)$$\mu =1+\frac{H_{0r}+j\;\alpha \;\hat{w}}{H_{0r}^{2}\;–\;{\hat{w}}^{2}+2\;j\;\alpha \;H_{0r}\;\hat{w}}$$
(7)$$\kappa =\frac{\hat{w}}{H_{0r}^{2}\;–\;{\hat{w}}^{2}+2\;j\;\alpha \;H_{0r}\;\hat{w}}$$
where: $H_{0r}=H_{0}/M_{S}$, $\hat{w}=\hat{f}/f_{m}$, $f_{m}=\gamma M_{S}$, $H_{0}$ is the static magnetic field in the sample (the internal static magnetic field), $M_{S}$ is the saturation magnetization, α is a Gilbert damping factor, γ=35.217 MHz/(kA/m), and $\hat{f}$ is the complex frequency. 

The imaginary part of frequency describes the time dependence of the electromagnetic fields (free oscillations) for a microwave resonator containing a ferromagnetic sample characterized by a permeability tensor (5). The relaxation time, $\tau =1/\left(\alpha \gamma H_{0}\right)$, and the ferromagnetic resonance linewidth, $\Delta H=2\alpha H_{0}=2/\left(\gamma \tau \right)$, are alternative parameters to the Gilbert damping factor for description of losses in ferromagnetic material.

It should be mentioned that expressions (6) and (7) are valid if the ferromagnetic medium is fully magnetized i.e., when $H_{0}>M_{S}$ when the domain structure in the ferromagnetic material vanishes.

Permeability of ferromagnetic material can be considered as a scalar quantity (8) for circularly polarized electromagnetic (EM) fields, orthogonal to the static magnetic field bias $H_{0}$ [2].
(8)$$\mu_{r,l}=\mu \pm \kappa$$
where μ and κ are the diagonal and off-diagonal components of the permeability tensor, respectively. Plus (minus) sign in Equation (8) corresponds to the clockwise (counter clockwise) polarization of the EM field. μ, κ and $\mu_{r}=\mu +\kappa$
exhibit a resonance character. Ferromagnetic resonance frequency $f_{FMR}$ is defined as the frequency for which the denominator in expression (6) or (7) approaches a minimum. This takes place when $f_{FMR}=\gamma H_{0}$ or $w=H_{0r}$. It should be noted that the real parts of μ and $\mu_{r}$ can be negative.

Permeability tenor becomes diagonal, and it is called the initial permeability for ferromagnetic medium in the completely demagnetized state. The initial permeability model, which is valid above the FMR frequency, has been developed by Schlömann. According to his model, the initial permeability can be written as follows [3].
(9)$$\mu_{d}=\frac{2}{3}{\left(\frac{w^{2}-{\left(H_{ar}+1\right)}^{2}}{w^{2}-H_{ar}^{2}\;}\right)}^{1/2}+\frac{1}{3}$$

$\mu_{d}$ values belong to the range 1/3 < $\mu_{d}$ < 1 and for high frequencies it monotonically approaches unity. Magnetic losses in this frequency region are small and decrease with frequency.

At frequencies lower than FMR values of the real part of the initial permeability are larger than unity, and their imaginary parts are also large which makes the ferromagnetic medium absorb the EM radiation. At this frequency range, magnetic domain wall resonances have a significant influence on the initial permeability dependence on the frequency.

### 2.3. Dispersion

Permittivity, permeability, and conductivity tensor components depend on a frequency which is known as dispersion. Dielectric relaxation is the relaxation response of a dipolar dielectric medium to an external EM field. It describes permittivity as a function of frequency (for ideal systems by the Debye equation). At infrared and optical frequencies, atomic and electronic resonances determine the permittivity response versus the frequency. Both Debye and resonance models predict local maxima of the imaginary part of permittivity at frequency ranges corresponding to the maximum slopes of the real part of permittivity versus frequency. For dielectric mixtures, the Maxwell–Wagner polarization accounts for the charge accumulation at the two-material interface, based on the difference of charge carrier relaxation times in these two materials. A full description of various dielectric dispersion mechanisms can be found in the famous A. von Hippel textbook [4]. The real part of permittivity in dielectric mixtures and dipolar dielectrics depend on the frequency, and associated dielectric losses are observed at a microwave frequency range. Typical dielectrics of this kind are dipolar liquids (e.g., alcohols and water). For most solid dielectrics, including common plastics dielectric minerals and pure (low doped) semiconductors, the real part of permittivity is almost constant at the microwave frequency range. However, the imaginary parts of permittivity for such materials usually vary with frequency.

Typical magnetic materials that are used at the microwave frequency range are ferrites and garnets. Being insulators, they do not introduce additional conductor losses. Ferrites and garnets exhibit significant dispersion of the initial permeability at a frequency range from a few hundred Hz to about 15 GHz. For hexagonal ferrites, the upper-frequency limit would be even higher. Dispersion of the initial permeability for these materials is associated with both the domain wall motion and ferromagnetic resonance in the magnetic anisotropy field of the sample under test.

Conductivity dispersion is also observed in some materials. The complex permittivity of metals exhibits negative values at optical frequencies. It can be described by the Drude model: (10)$$\varepsilon_{r}=\varepsilon_{r}^{\prime }-j\varepsilon_{r}^{\prime \prime }=\varepsilon_{\infty }-\frac{\omega_{p}^{2}}{\hat{\omega }{}^{2}-j\hat{\omega }\Gamma }$$
where $\omega_{p}$ is angular plasma frequency, $\varepsilon_{\infty }$ is permittivity at an infinite frequency, and Γ is a damping factor. Plasma frequency depends on the square root of the density of the free charge carriers.

Superconductors interact with electromagnetic fields differently than metals. The conductivity of superconductors at microwave frequencies can be characterized by the two-fluid model. According to this model, the conductivity of an isotropic superconductor is the complex-valued number: $\sigma (\omega )=\sigma_{1}(\omega )-j\sigma_{2}(\omega )$. The imaginary part $\sigma_{2}\left(\omega \right)$ is related to superconductivity, and converges to infinity as the frequency approaches zero. The real part of conductivity $\sigma_{1}\left(\omega \right)$ is related to all loss mechanisms that are present in the superconductor, including normal conductivity. Substituting the complex conductivity into: $\varepsilon =\left(\varepsilon_{d}-j\sigma /\omega \right)$ and neglecting $\varepsilon_{d}$, one obtains the following expression for the complex permittivity of a superconductor:(11)$$\varepsilon =\varepsilon_{0}\left(-\frac{\sigma_{2}}{\omega \varepsilon_{0}}-j\frac{\sigma_{1}}{\omega \varepsilon_{0}}\right)$$

As is seen in (11), superconductors exhibit large negative permittivity related to $\sigma_{2}$, known as the kinetic inductance. As a consequence of the negative permittivity, the EM fields in a medium are attenuated. Penetration depth for a lossless superconductor (the London penetration depth $\lambda_{L}$), can be calculated as (12).
(12)$$\lambda ={\left(\frac{1}{\omega \mu_{0}\sigma_{2}}\right)}^{0.5}=\lambda_{L}$$

The London penetration depth value is considered as a frequency-independent quantity.

## 3. Microwave Measurement Techniques

Measurement techniques of the complex permittivity and the complex permeability at microwave frequencies can be divided into transmission/reflection methods and resonance methods. Transmission/reflection methods are usually used for broad frequency band measurements of medium and high loss dielectric and magnetic materials, while the resonance ones are for single-frequency measurements of materials that exhibit arbitrary losses. Some resonance techniques employ not just one but several modes, allowing measurements in a broad but sparse frequency spectrum.

In transmission/reflection methods, the complex permittivity and the complex permeability of isotropic materials are determined from the measured transmission and reflection coefficients of the measurement cell containing the sample under test. For nonmagnetic samples, a measurement of a single coefficient (reflection or transmission) is sufficient to evaluate the complex permittivity value. In resonance methods, measurements of the resonance frequency and Q-factor of a resonator containing sample under test are necessary for the complex permittivity determination. It should be mentioned that the size of samples measured at microwave frequencies is typically comparable to the wavelength, so the numerical solutions of Maxwell’s equations have to be employed to find the relationship between the measured and the calculated quantities.

These days, measurements of complex-valued transmission and reflection coefficients as well as measurements of the resonance frequencies and Q-factors at microwave frequencies are performed employing a vector network analyzer (VNA). Appropriate measurement cells (transmission/reflection ones or resonators) are connected to VNA through flexible coaxial cables.

### 3.1. Transmission/Reflection Cells

Several different measurement cells are employed for transmission/reflection measurements [5,6,7,8,9,10,11,12,13,14,15,16,17,18,19]. Some of them are schematically shown in Figure 1a–d. For cells shown in Figure 1a,b, the sample under test occupies the whole cross-section of a coaxial line or waveguide. Measured quantities of a cell containing a sample are complex-valued scattering matrix (*S*–matrix) parameters. For the reciprocal medium, two of them, $S_{11}$ (reflection coefficient) and $S_{21}$ (transmission coefficient) have to be measured at a certain frequency range. For isotropic materials, from the measured $S_{11}$ and $S_{21}$ values, the complex permittivity and the complex permeability can be simultaneously determined [5,8] as functions of frequency. The open-ended coaxial probe (Figure 1c) and the open-ended waveguide cell (Figure 1d) are used for the determination of the complex permittivity of isotropic nonmagnetic materials from measured $S_{11}$ values versus frequency [12,13,14,15,16,17,18]. For large permittivity materials, samples should be small, as is depicted in Figure 2, to avoid resonances in the sample. One of the main sources of measurement uncertainties in transmission/reflection measurement cells is air gaps between the sample and metal walls of the coaxial line or waveguide. To minimize the influence of air gaps, the surfaces of the samples that directly touch surfaces of the cell should be metalized. The main advantages of transmission/reflection measurement techniques are: relatively broad-band frequency coverage (about 30:1 for coaxial transmission line method [19]), the uncertainties for the real part of permittivity are typically better than ±1% [19], especially for measurements of medium and high loss liquids when the air gap problem does not exist. Their main disadvantages of transmission/reflection methods are that for solid specimens, significant measurement errors occur from the presence of air gaps between the sample and metal parts of the measurements cell, and their resolution of loss tangent measurements is limited (typically to ±0.01) [19].

### 3.2. Resonance Cells

Resonance techniques can be divided into two groups. For the first group, a sample under test is measured in some kind of microwave resonator, such as a metal cavity, open resonator, or a more complicated resonance structure consisting of metal and dielectric elements. Usually, for such resonators, the electric energy in the sample under test is much smaller than the electric energy in the whole resonance structure. For the second group, the sample under test accumulates the dominant part of the electric energy and creates some kind of “dielectric resonator”. Resonators of both groups can operate on different modes. Typically, only one mode is employed but sometimes few modes belonging to a certain mode family are used. Measured quantities are the resonance frequencies and the Q-factors for the modes of interest. The relationship between the measured quantities and the complex permittivity of the sample requires solutions to Maxwell’s equations for the resonance structure which is used for the measurements. Solutions in a close analytic form (transcendental equations) are known only for simple spherical, cylindrical, or rectangular geometries. For more complicated structures, numerical methods of electrodynamic analysis have to be employed. The general form of the relationship between the complex permittivity of the sample and the measured quantities has the following form [20]:(13)$$F(\varepsilon_{r},\omega )=0$$

In (13), ω denotes the angular complex frequency for a specific mode of free oscillations, Re(ω)=2πf, *f*—the resonance frequency, $Im\left(\omega \right)=Re\left(\omega \right)/(2Q_{d})$, $Q_{d}$ denotes the *Q*-factor depending on the dielectric and radiation losses in the resonant structure. *Q*-factor that is depending on the metal wall losses has to be additionally computed. More details on this topic can be found in [20]. For a long time, metal cavities having cylindrical or rectangular shapes have been used for the complex permittivity measurements [21,22]. With the development of computers and numerical methods of electrodynamics, more complicated resonance structures are used that employ the so-called “dielectric resonator” techniques [23,24,25,26,27,28,29,30,31,32,33,34,35,36,37,38] which are shown in Figure 3a–c and Figure 4a–d. For such structures, numerical electrodynamic methods have to be used to find the relationship between the complex resonance frequency and the complex permittivity of the sample. Probably one of the most frequently used resonators for non-destructive measurements of printed wiring boards (PWB) dielectric substrates are the split post dielectric resonators (SPDR) depicted in Figure 4a [30,31,32]. They are also used for the contactless measurements of high resistivity semiconductors with $10^{2}<\rho <10^{5}$ (Ωcm). The single post dielectric resonators (SiPDR) shown in Figure 4b [33,34] are used for the contactless resistivity measurements of semiconducting wafers in the resistivity range $10^{-6}<\rho <10^{3}$ (Ωcm) and for the sheet resistance measurements of thin conducting films deposited on a dielectric substrate, including such materials as graphene or conducting polymers. For both the single post and the split post dielectric resonators, the electric field is circumferential with a radial distribution as shown in Figure 4c (for resonator operating at 5 GHz). It should be mentioned that applications of the split post and the single post dielectric resonators, due to their relatively small size, enable localized resistivity measurements on relatively small areas of larger semiconducting wafers (resistivity mapping). The size of resonators become smaller if they operate at higher frequency bands (dimensional scaling is inversely proportional to frequency), allowing for the probing of small areas on the measured wafers. Sapphire rod resonator shown in Figure 4d allows very accurate measurements of the surface resistance, and sometimes also the surface reactance of superconductors, such as yttrium barium copper oxide (YBCO), due to the extremely low loss of sapphire at cryogenic temperatures. At millimeter-wave frequencies open Fabry–Perot type resonators are used [39,40,41,42,43,44,45] for measurements of flat low loss dielectric samples, shown in Figure 5a,b.

## 4. Microwave Measurements of Dielectrics

The complex permittivity measurements of dielectrics are probably the most frequent tests of materials performed at microwave frequencies. Measurement technique predominantly depends on the dielectric losses. Medium and high loss dielectrics having $\tan\delta >10^{-2}$, especially liquids, are typically measured employing various transmission/reflection cells presented in Figure 2. Low loss dielectrics are measured using resonance techniques. The most accurate measurements of the complex permittivity of low loss dielectrics can be performed employing the “dielectric resonator” technique, where the dominant part of the electric energy is concentrated in the sample under test. One of the first methods of this kind was the Hakki–Coleman technique (Figure 3a), whereby a cylindrical rod sample was situated between two large conducting plates [23]. The mode which is used in the Hakki–Coleman measurements is the $TE_{011}$ one, having only an azimuthal electric field component which is tangential to the surfaces of the sample. Electromagnetic fields decay outside the sample if the sample has a sufficiently large aspect ratio (diameter to height). This is important for low permittivity samples (such as polymers) because the minimum aspect ratio decreases with permittivity. Exact solutions of Maxwell’s equations are available for the Hakki–Coleman resonator, but losses in metal plates (which are difficult to be measured) constitute a significant part of the overall resonator losses. Therefore, the loss tangent resolution and accuracy of this technique are not so good for the $TE_{01\delta }$ mode dielectric resonator structure shown in Figure 3c [28]. The only disadvantage of the $TE_{01\delta }$ mode dielectric resonator is that numerical methods of electrodynamics have to be used to determine the complex permittivity. The same resonance structure Figure 3c can be employed for measurements of higher-order quasi-$TE_{0np}$ [29] and whispering gallery modes [37,38] in the sample at higher frequencies. Using the whispering gallery modes belonging to quasi-TE and quasi-TM families, one can determine two permittivity components for oriented uniaxially anisotropic samples. For higher-order whispering gallery modes, the electromagnetic energy is predominantly concentrated in the sample near its lateral surface. For this reason, conductor losses in the metal shield are negligibly small and the dielectric loss tangent of extremely low loss dielectrics can be determined. Using the whispering gallery mode technique, it was possible to measure the dielectric loss tangent of sapphire as being equal to about $10^{-10}$ at the liquid helium temperature (at 10 GHz) [37].

Employing “dielectric resonator” techniques (using the whispering gallery and quasi-$TE_{0mn}$ modes) several low loss single-crystal dielectrics have been measured [38,46,47,48,49,50,51,52,53,54,55,56,57]. In Table 1, results of room temperature permittivity and the thermal coefficient of permittivity measurements using these techniques for several single-crystal dielectrics are presented. The first five materials in Table 1 were uniaxially anisotropic dielectrics, whereby two whispering gallery modes method was employed to determine the two permittivity tensor components. The other dielectrics were measured employing a single-mode (the whispering gallery or quasi-$TE_{0np}$).

Resonance techniques are also applicable for the complex permittivity measurements of high permittivity lossy dielectrics (ferroelectrics). To maintain a sufficiently large Q-factor of the resonance cell containing ferroelectric sample, it is necessary to significantly reduce the amount of the electric energy filling factor (ratio of the electric energy stored in the sample to the total electric energy in the resonance cell). One of the resonators that allows such measurements on small cylindrical rod samples is presented in Figure 3b. Measurements on a PbMg (1/3) Nb (2/3) O_3_ (PMN) ceramic sample having a diameter of 0.76 mm have been performed at a frequency of about 0.88 GHz, employing $TE_{0mn}$ modes [25]. Results of the complex permittivity measurements versus temperature are shown in Figure 6. As it has been mentioned earlier for these modes, the electric field is circumferential which practically mitigates the air gap problem. At millimeter-wave frequency dimensions of resonance cells containing dielectric resonators, as shown in Figure 3a–c, become small and in this frequency range open Fabry–Perot type resonators are still used [39,40,41,42,43,44,45]. The recently improved electrodynamic theory of Fabry–Perot resonator and the new fully automatic measurement system have been developed [45], that allow accurate measurements of the complex permittivity of low and medium loss dielectrics at several frequencies in the range from 20 GHz to 110 GHz.

Most of the methods described in this section, except whispering gallery mode methods, are intended for measurements of the in-plane permittivity in the sample. Because some materials of practical use, such as PWB substrates can exhibit anisotropy, complementary methods have to be used to measure permittivity components perpendicular to the substrate. Several resonance methods allow such measurements that employ TM and quasi-TM modes. An overview of such methods has been presented by Dankov [22].

By summarizing the permittivity measurements employing resonance cells, we can formulate the following recommendations.

The most accurate techniques for measurements of low loss dielectric having $\tan\delta <10^{-2}$ are the “dielectric resonator” methods, operating on whispering gallery or quasi-$TE_{0np}$ modes. Measurements of the two permittivity components of uniaxially anisotropic crystals are possible by employing two different whispering gallery modes. Dielectric resonator techniques can be used for measurements of materials that have arbitrary permittivity and extremely low losses. They are especially useful for measurements of the thermal coefficient of permittivity. The frequency limits for these methods are related to the size of the sample, its permittivity, and the mode of operation because the sample under test creates a “dielectric resonator”. Samples for the dielectric resonator methods have to be machined in the form of discs or cylinders, with dimensions in the order of few millimeters.

Non-destructive measurements of the in-plane permittivity for dielectric substrates can be conveniently performed by employing SPDR’s at a frequency range from 1 GHz to 20 GHz, and Fabry–Perot resonators at a frequency range from 20 GHz to 110 GHz. Loss tangent resolution for these methods is the order of $2\times 10^{-5}$ and the uncertainty for the real part of permittivity is typically better than ±1% for well-machined samples. High permittivity samples have to be sufficiently thin, especially at millimeter wave frequencies to avoid resonances in the sample.

Measurements of both the real part and the imaginary part of permittivity on lossy dielectrics, through employing any technique, are possible only for materials that have the imaginary part of permittivity of which is smaller, or the same order of magnitude as the real part, otherwise only the imaginary part of permittivity can be determined. When the imaginary part of permittivity is much greater than the real part, the wave impedance and the propagation constant practically depend only on the imaginary part of permittivity. For this reason, it is impossible to measure the real part of permittivity for medium and heavily doped semiconductors or metals.

## 5. Microwave Measurements on Semiconductors, Conductors, and Superconductors

For isotropic, nonmagnetic semiconducting or conducting medium its complex permittivity can be represented as:(14)$$\varepsilon =\varepsilon_{0}\varepsilon_{r}=\varepsilon_{0}(\varepsilon_{r}^{\prime }-j\varepsilon_{rd}^{\prime \prime }-j\frac{\sigma }{\omega \varepsilon_{0}})=\varepsilon_{0}\varepsilon_{r}^{\prime }(1-j\tan\delta_{eff})$$
where: $\varepsilon_{r}^{\prime }$—the real part ($\varepsilon_{rd}^{\prime \prime }$—the imaginary part) of the relative permittivity associated with the dielectric polarization losses, σ-conductivity, $\varepsilon_{eff}^{\prime \prime }=\varepsilon_{rd}^{\prime \prime }+\frac{\sigma }{\omega \varepsilon_{0}}$, and $\tan\delta_{eff}=\tan\delta_{d}+\frac{\sigma }{\omega \varepsilon_{0}\varepsilon_{r}^{\prime }}$.

Measurement of the imaginary part of permittivity at a single frequency provides information on the total losses that include conductivity terms. The dielectric polarization loss term must be known (or much smaller than the conductivity term) to determine conductivity. The higher the frequency, the smaller the effective loss tangent term which depends on the conductor losses i.e., $\sigma /\left(\omega \varepsilon_{0}\varepsilon_{r}^{\prime }\right)$. For single-crystal semiconductors, the dielectric polarization loss tangent term at microwave frequencies and room temperatures is the order of 10^−4^, so the maximum resistivity value when the conductor term is dominant is the order of 10^5^ Ωcm.

As is seen in Figure 7, semiconductors that exhibit resistivity in the range from 10^3^ to 10^5^ Ωcm at the microwave frequency range have the effective dielectric loss tangent in the range 10^−2^ to 10^−4^, and behave similar to low loss dielectrics. For this reason, bulk cylindrical rod samples made of such semiconductors can be conveniently measured by employing the $TE_{01\delta }$ mode dielectric resonator technique [28]. Such a technique is the most accurate for the determination of the complex permittivity. Several have been measured by this technique versus temperature and frequency (employing higher-order modes excited in one sample). Results of high resistivity semiconductor measurements, including Si [58,59], GaAs [60], GaP [60], SiC [61], GaN [62] have already been published. Because the term of the effective dielectric loss tangent depending on conductivity decreases with frequency in a well-known manner (8), the combination of frequency and temperature measurements can separate the dielectric and the conducting terms, even if they are comparable. In Figure 8a, the results of the permittivity measurements of a silicon sample having a nominal resistivity of about 13 kΩcm at room temperature are presented [58]. As it is seen at temperatures above 200 K, permittivity increases linearly with temperature, while at temperatures below 10 K, it is almost temperature independent. Such behavior is typical of most high resistivity semiconductors and low loss dielectrics. In Figure 8b, the results of resistivity measurements of high resistivity silicon samples are presented [59]. Red solid points correspond to the as-grown float zone (FZ) silicon sample, having a nominal resistivity of about 85 kΩcm at room temperature. Open blue circles correspond to FZ proton irradiated sample. Green squares correspond to DC resistivity measurements performed on another FZ proton irradiated sample. It has been shown [59] that high resistivity silicon irradiated with a sufficiently large dose of high energy protons or neutrons behaves similar to that of the intrinsic silicon. Irradiation introduces defects that act as traps for electrons, and holes are created by residual doping. More information on measurements of these silicon samples can be found in the original paper [59]. Results of the loss tangent determination depending on the dielectric losses for bulk samples of semi-insulating Si [58,59], GaAs [60], GaP [60], and GaN (type 1) [62] are shown in Figure 8c, while results of the effective loss tangent measurements versus temperature on two samples of GaN are shown in Figure 8d. Measurements of the effective dielectric loss tangent of GaN have been performed on samples obtained with the ammonothermal growth. Two samples type 1 were doped with transition metal ions as deep acceptors, while sample type 2 was doped with Mg ions as shallow acceptors.

Two samples of type 1 had the following dimensions, d = 7.84 mm, h = 1.74 mm and d = 8.20 mm, h = 1.67 mm. The sample type 2 had dimensions d = 7.84 mm, h = 1.74 mm. It is seen in Figure 8c that samples GaAs, GaP, and GaN type 1 behave like ordinary dielectrics. For sample GaN, type 2, one can observe exponential growth of the effective dielectric loss tangent versus temperature above 400 K (Figure 8d). At temperatures above 400 K, GaN sample type 2 is no longer compensated and thus semi-insulating.

$TE_{01\delta }$ mode dielectric resonator technique can be modified to extend the measurement range of semiconductors towards lower resistivities, by employing smaller samples placed at the center of the cylindrical dielectric resonator (Figure 6), similarly as in measurements of ferroelectrics.

Bulk semiconductor samples are rarely manufactured for industrial applications, and most of the semiconductors are available in a form of round wafers with a thickness in the range from 0.1 mm to 1.0 mm. Furthermore, the resistivity of commercially available semiconductor wafers would extend over several orders of magnitude. As it has been mentioned earlier, the resistivity of such wafers can be conveniently determined using the single post dielectric resonator technique [33,34]. Determination of resistivity is based on simultaneous measurements of the resonance frequency and the Q-factor of a single post dielectric resonator with the wafer under test.

As it is seen in Figure 9a,b, measurements of the resonance frequency shift and Q-factor allow for the unique determination of resistivity over nine orders of magnitude. Single post dielectric resonators operating at 5 GHz are commercially available and allow measurements of resistivity in a very short time. It should be also mentioned that the existence of thin dielectric layers on the wafer (such as the residual layer of SiO2 on silicon) does not influence the resistivity measurement results. A single post dielectric resonator can be also used for the sheet resistance measurements of thin conducting films including graphene [63].

Microwave measurements of resistivity/conductivity of well-conducting metals (Ag, Cu, Al) and superconductors require resonance cells with very small “parasitic losses”, i.e., losses in all other parts of the resonance structure except the sample. One of the best resonators of this kind is the sapphire rod dielectric resonator, operating on the $TE_{011}$ mode (Figure 4d) [36]. The dielectric loss tangent of sapphire at temperatures T < 80 K is < 10^−7^. Due to the evanescent character of the electromagnetic fields outside the sapphire rod, conductor losses in the lateral surface of the structure are very small and calculable, so the measured Q-factor of such resonator containing samples under test predominantly depends on the losses in samples under test. Two identical samples are required to ensure the highest sensitivity of measurements. The sheet resistance of the samples can be determined from the Q-factor measurements. For superconducting films that are much thinner than the penetration depth, it is possible to measure the resonance frequency shift related to such samples with respect to the resonance frequency of resonator with copper metal plates. In such cases, the complex surface impedance and the complex conductivity can be measured [36]. Results of the complex conductivity measurements of YBCO films having a thickness of 30 nm are shown in Figure 10a, and the effective penetration depth evaluated from these measurements is shown in Figure 10b. At low temperatures below the critical temperature of YBCO, the effective penetration depth predominantly depends on the superconductivity of the sample, and above this temperature it is simply the skin depth, depending on its real part of conductivity.

At room temperatures, a sapphire rod resonator can be used to measure the conductivity of electrolytic layers of metals deposited on another bulk metal. It has also been used to measure the effective conductivity of PWB copper metallization through the dielectric substrate [64]. This conductivity can be a few times smaller than the conductivity of bulk copper, due to the surface roughness. In Figure 11, the resistivity ranges for contactless microwave and RF measurements methods are presented. Detailed descriptions of various contactless methods of resistivity measurements have been presented in [34].

## 6. Microwave Measurements of Ferrites

Ferrites are commonly used at microwave frequencies in magnetically tunable devices, power limiters, oscillators, circulators, isolators, as well as in microwave absorbers. For most microwave devices, magnetic losses should be small, except for microwave absorbers. Because the initial permeability of ferrites exhibits significant dispersion at microwave frequencies, its imaginary part would vary by several orders of magnitude. At the lower part of the frequency spectrum, both the real part and the imaginary part of permeability are large, while at frequencies $f>\gamma M_{S}$, the real part of permeability is close to unity and the imaginary part is small. For this reason, different measurement techniques are used at these two frequency ranges. Measurements at $f<\gamma M_{S}$ are usually performed employing the transmission/reflection methods, while at $f>\gamma M_{S}$ employing resonance techniques. Typical results of the initial permeability measurements obtained from coaxial transmission/reflection cell (Figure 3a) data for the polycrystalline yttrium iron garnet (YIG) sample are shown in Figure 12 [65]. It is seen that at frequencies at $f<10^{8}$ Hz, yttrium iron garnet behaves like an absorbing material. At a frequency range from 0.6 GHz to 1.5 GHz the real part of the initial permeability is negative, and at higher frequencies converges to unity. At frequencies that are larger than the ferromagnetic resonance frequency, $f_{FMR}=\gamma M_{S}\approx 4.93$ (GHz) for YIG, the imaginary part of the initial permeability is very small and cannot be accurately measured by employing the transmission/reflection method. One of the most accurate methods of the initial permeability measurements at frequencies $f>\gamma M_{S}$ is a cylindrical ring dielectric resonator [27]. Employing several ring resonators having different diameters and permittivity allows measurements on a relatively broad but sparse frequency spectrum. Such measurements have been performed on several microwave ferrites at frequencies from $\gamma M_{S}$ up to 25 GHz [27]. Those measurements qualitatively confirm Schlömann’s model of the initial permeability. However, it should be mentioned that the initial permeability depends on the magnetic domain distribution within the sample under test.

For thin samples with relatively large areas, as microwave substrates, magnetic domains are mostly oriented in the direction parallel to the surface of the sample. For this reason, the initial permeabilities measured in the direction parallel and perpendicular to the surface of the sample are different. This is seen in Figure 13a, where results of the initial permeability measurements on polycrystalline samples having a thickness of 0.383 mm are presented. Measurements have been performed by employing single post and split post dielectric resonators. More details about those measurements can be found in the original paper [66].

It should also be mentioned that very accurate measurements of the initial permeability at a frequency that is sufficiently higher than $\gamma M_{S}$, when the imaginary part of the initial part is the order of $10^{-3}$, can be performed by the Courtney method [24], in which a rod shape sample creates a dielectric/magnetic resonator (Figure 4a). Courtney’s method is just a modification of the Hakki–Coleman dielectric resonator method. Measurements are performed twice: with very strong static magnetic field bias (the order of 2−3 T) and without the bias. It is assumed that the initial permeability is equal to unity in the first measurements, so the permittivity of the sample can be determined. Having known the permittivity of the sample, permeability is evaluated from the second measurement.

The determination of the three permeability tensor components by employing resonance measurements, preferably performed on one sample, requires the use of three different modes [26]. It can be accomplished by employing two cylindrical dielectric resonators of the same height but having different external diameters, as shown in Figure 14a. The larger resonator operates on the $TE_{011}$ mode, and the smaller on the $HE_{111}$ mode. Without any bias, $HE_{111}$ mode is doubly degenerated. In the presence of a static magnetic field bias mode, degeneracy vanishes and two circularly polarized $HE_{111}^{-}$, $HE_{111}^{+}$ modes appear. At zero bias, resonant frequencies of the degenerated $HE_{111}$ mode and the $TE_{011}$ mode depend only on the initial permeability and the scalar initial permittivity, so these quantities can be determined from the appropriate transcendental equations. When permittivity is known, all three permeability tensor components can be determined at a fixed bias from the three measured frequencies. Results of measurements of all three permeability tensor components on polycrystalline YIG samples are shown in Figure 14b. Measurement procedures have been described in detail in [26,66,67].

The dependence of permeability tensor components on the static magnetic field very well agrees with the theoretical model (5), (6), (7) for $H_{ext}$ > 50 kA/m, when ferrite material is practically saturated. For $H_{ext}$ < 50 kA/m, measurement results create a reliable database for the permeability tensor of the sample.

Single crystal YIG spherical samples behave similar to tunable dielectric resonators, and are commonly used in YIG filters and oscillators. The dominant mode occurring in a spherical sample subjected to a static magnetic field bias is referred to in the literature as the mode of uniform precession, which occurs when the effective permeability of infinitesimally small sample $\mu_{r}$ (8) is equal to −2. Recently, it has been shown on the ground of the electrodynamic, the mode of uniform precession corresponds to the mode having properties of the magnetic plasmon resonance [68]. Through employing the electrodynamic theory, the accurate method of measurements of the ferromagnetic resonance linewidth has been proposed which is not limited by the size of the sample [69]. More details about measurements of ferrites can be found in the overview paper [66]. It should be underlined that for sufficiently strong static magnetic field bias, the permeability tensor of ferrites with a weak magnetic anisotropy is well characterized by the saturation magnetization (static parameter) and the ferromagnetic linewidth (microwave parameter).

## 7. Summary

This paper reviews only a fraction of techniques intended for the measurement of material properties at microwave frequencies. Techniques that have been described in the vast number of publications employ various resonance and non-resonance cells, based on stripline, microstrip, coplanar waveguide technologies, and metal cavities of a different kind. The general feature of all techniques is that the resolution of the loss tangent measurements is associated with the presence of parasitic losses in the measurement cell. They must be sufficiently small and calculable to determine the imaginary parts of the permittivity/permeability for very low loss materials. For the whispering gallery modes, parasitic losses are negligibly small, so by employing these modes it has been possible to measure the dielectric loss tangent values in the order of $10^{-10}$. Simultaneous permittivity and permeability determination for isotropic magnetic materials and the determination of all dielectric or magnetic tensor components for anisotropic materials generally requires measurements of at least the same number of independent parameters as the number of unknowns. As it has been mentioned earlier that the rigorous determination of permittivity (permeability) at microwave frequencies requires rigorous numerical electrodynamic modeling of measurement cells containing the samples under test. This is one of the most challenging tasks for modern microwave metrology.

## Figures and Tables

**Figure 1 materials-14-05097-f001:**
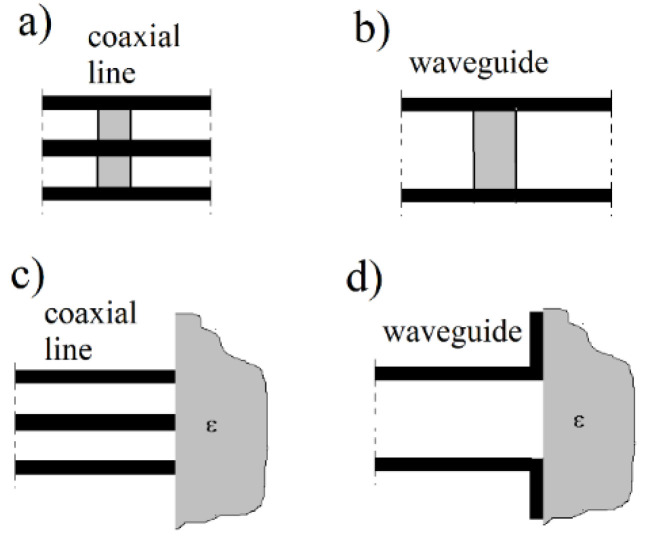
(**a**,**b**) Transmission/reflection and (**c**,**d**) reflection measurement cells.

**Figure 2 materials-14-05097-f002:**
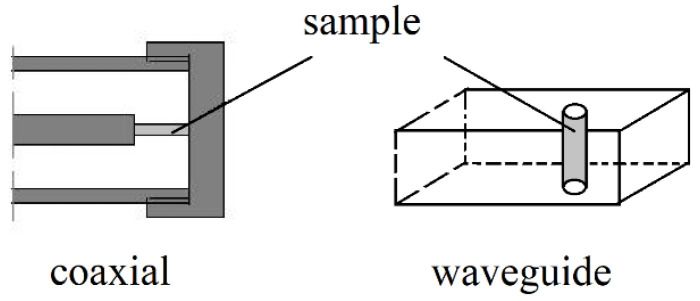
Reflection cells intended for measurements of high permittivity materials.

**Figure 3 materials-14-05097-f003:**
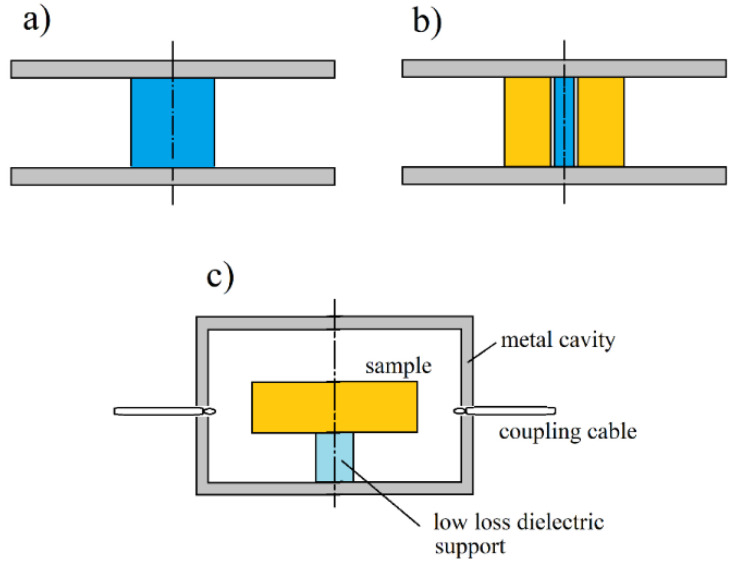
$TE_{011}$ mode dielectric resonators (**a**) Hakki–Coleman cell [23] for measurements of low loss dielectrics, (**b**) cylindrical dielectric resonator for measurements of high permittivity high loss samples [25] and measurements of the initial permeability of ferrites [27], (**c**) $TE_{01\delta }$ mode dielectric resonator inside cylindrical metal shield [28]. The same resonance structure can be used for measurements of higher-order quasi $TE_{0np}$ [29] and whispering gallery modes [37,38] at higher frequencies.

**Figure 4 materials-14-05097-f004:**
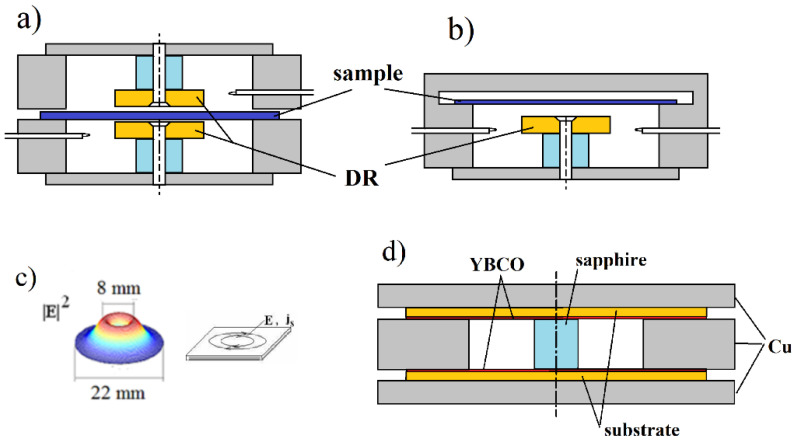
(**a**) Split post dielectric resonator [31,32] for measurements of low and medium loss dielectrics and high resistivity semiconductors, (**b**) single post dielectric resonator [33,34] for measurements of resistivity and the sheet resistance of semiconductors and conducting films, (**c**) electric field distribution in the single post dielectric resonator operating at 5 GHz, (**d**) sapphire rod dielectric resonator, operating on the $TE_{011}$ mode, for measurements of conductivity of metals and superconductors [35,36].

**Figure 5 materials-14-05097-f005:**
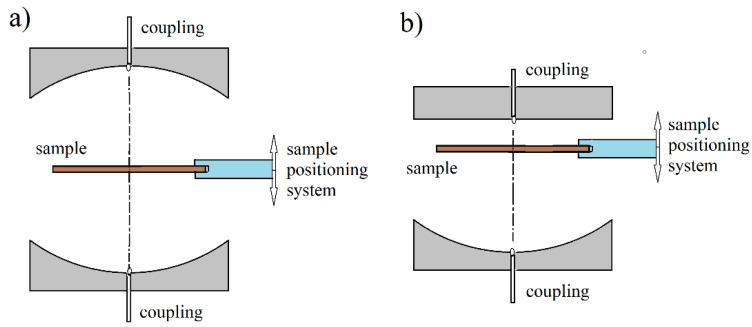
Fabry–Perot open resonators for measurements of low loss dielectrics at millimeter-wave frequencies [45]. (**a**) resonator with double concave reflectors, (**b**) resonator with plane and concave reflectors.

**Figure 6 materials-14-05097-f006:**
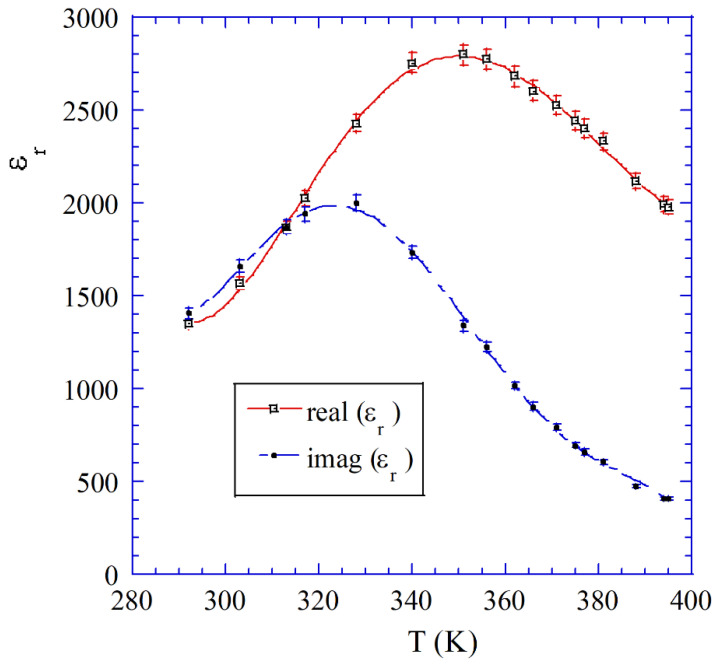
Real imaginary parts of permittivity of PMN sample versus temperature (data taken from [25]).

**Figure 7 materials-14-05097-f007:**
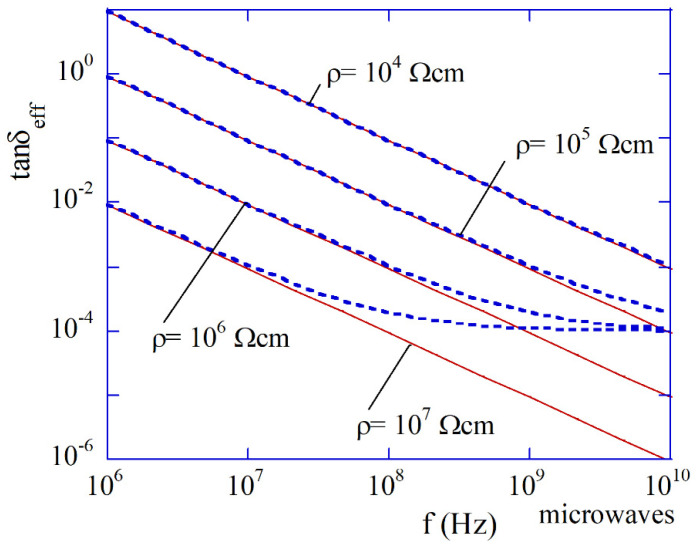
The effective loss tangents of high resistivity semiconductors as functions of frequency assuming dielectric polarization loss tangent term equal to 10^−4^. Solid lines denote terms depending on pure conductor losses i.e., $\sigma /\left(\omega \varepsilon_{0}\varepsilon_{r}^{\prime }\right)$.

**Figure 8 materials-14-05097-f008:**
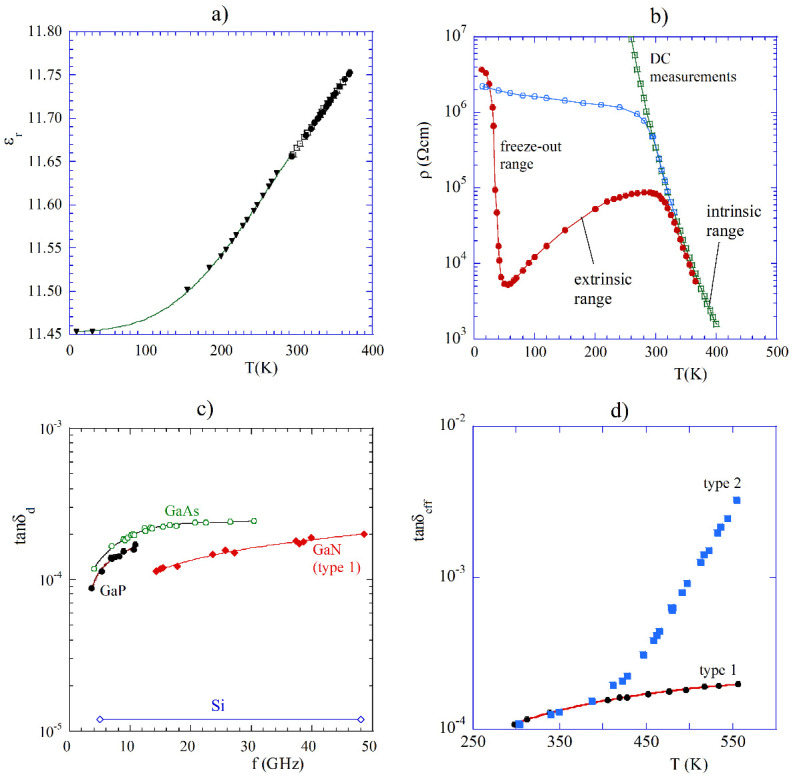
(**a**) Permittivity of high resistivity silicon sample having nominal resistivity of about 13 kΩcm at room temperature [58] © IEEE Publishing. Reproduced with permission. All rights reserved. (**b**) Resistivity of high resistivity silicon samples measured with $TE_{01\delta }$ mode dielectric resonator technique. Red solid points correspond to the as-grown float zone (FZ) silicon sample, having nominal resistivity of about 85 kΩcm at room temperature [59]. Open blue circles correspond to FZ proton irradiated sample. DC resistivity measurements (green squares) have been performed on another FZ proton irradiated sample. (**c**) The dielectric loss tangent, due to dielectric losses of semi-insulating semiconductors, at the temperature of 25 C. (**d**) The total dielectric loss tangent of two kinds of GaN samples as a function of temperature at a frequency about 15 GHz [62].

**Figure 9 materials-14-05097-f009:**
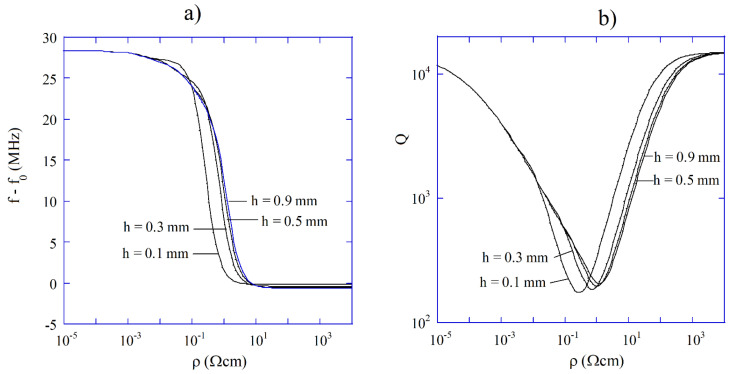
(**a**) Resonance frequency shift of 5 GHz single post dielectric resonator versus resistivity of semiconductor sample, (**b**) Q-factor of 5 GHz single post dielectric resonator versus resistivity of a semiconductor sample.

**Figure 10 materials-14-05097-f010:**
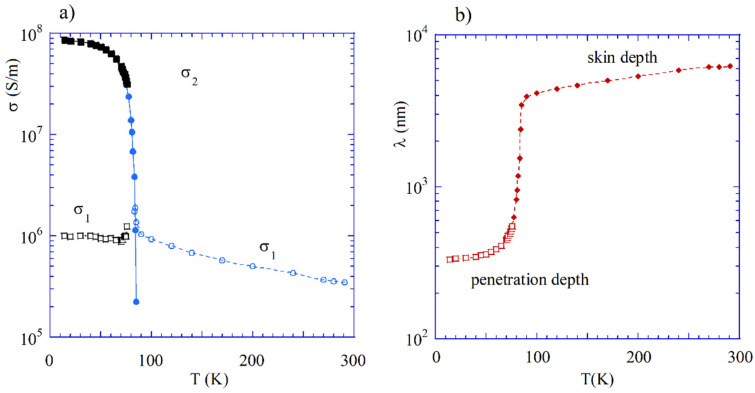
(**a**) Components of the complex conductivity of YBCO sample measured with sapphire rod resonator (at temperatures below 80 K) and single post dielectric resonator (above 80 K). (**b**) The effective penetration depth evolution of YBCO as a function of temperature, reflecting conductivity transition from the superconducting state to the normal state. © IEEE Publishing. Reproduced from [36] with permission. All rights reserved.

**Figure 11 materials-14-05097-f011:**
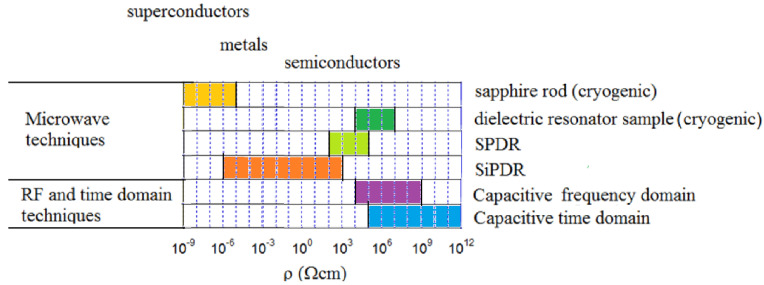
Resistivity ranges for contactless microwave and RF measurements methods [34]. SiPDR—single post dielectric resonator, SPDR—split post dielectric resonator.

**Figure 12 materials-14-05097-f012:**
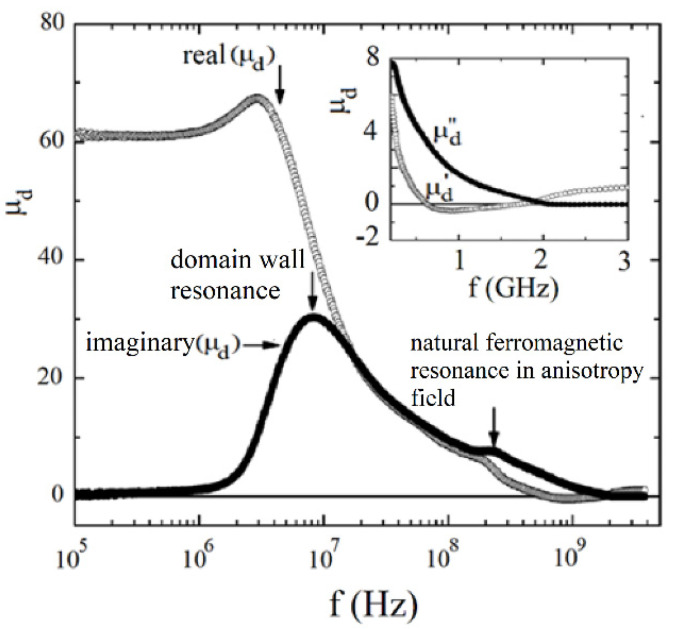
Measurement results of the initial permeability for polycrystalline YIG sample having a thickness of 1.0 mm, and employing coaxial line transmission/reflection cell with the outer diameter of 7 mm. © Reproduced from [65], with the permission of AIP Publishing.

**Figure 13 materials-14-05097-f013:**
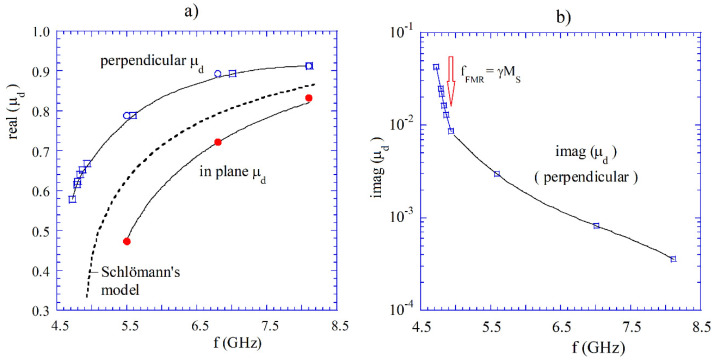
(**a**) The initial permeability of ceramic YIG substrates (thickness of 0.383 mm) measured with microwave field directions parallel and perpendicular to the surface of samples, employing split post and single post dielectric resonators [66]. (**b**) The imaginary part of the initial permeability perpendicular to the surface of the sample [66].

**Figure 14 materials-14-05097-f014:**
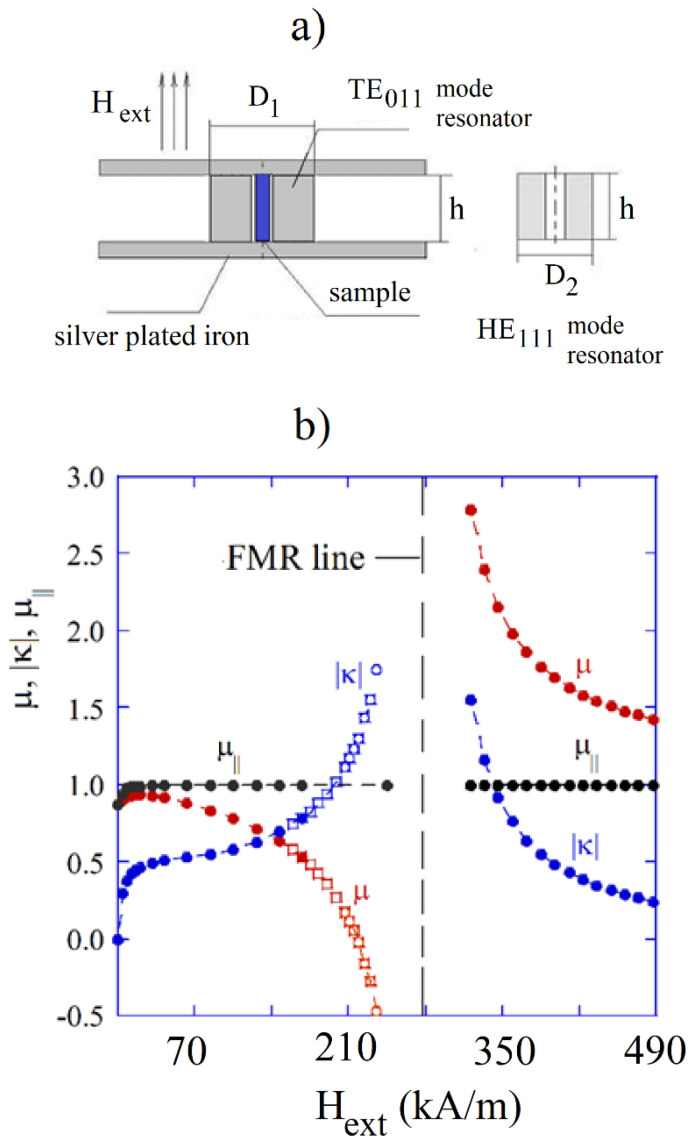
(**a**) Two dielectric resonators for measurements of permeability tensor. (**b**) Results of measurements of three permeability tensor components of polycrystalline YIG scaled to frequency 9.82 GHz, dots refer to the sample having a diameter of 2.00 mm, squares refer to the sample having d = 0.47 mm [67].

**Table 1 materials-14-05097-t001:** Permittivity of some dielectric single crystals at T = 300 K.

Material	$\varepsilon_{⊥}$	$\begin{array}{c} TK_{\varepsilon_{⊥}}\\ \left(\mathrm{ppm}/K\right) \end{array}$	$\varepsilon_{\|}$	$\begin{array}{c} TK_{\varepsilon_{\|}}\\ \left(\mathrm{ppm}/K\right) \end{array}$	f (GHz)	Resonance Type	Ref.
Sapphire (Al_2_O_3_)	9.395	+85	11.59	+121	21.5; 5–16	WGM	[38,51]
SrLaAlO_4_	16.85	+50	19.80	+115	12	WGM	[38]
Rutile (TiO_2_)	85.7	−760	163.2	−1200	2.5–5.5	WGM	[46]
Quartz (SiO_2_)	4.435	+9	4.64	+28.7	17	WGM	[38]
La_6_Ga_11_TaO_28_	18.15	+107	50.0	−810	3–10	WGM	[47]
YAG	10.60	+108	−−−	−−−−	20;	WGM	[38,54]
LiTaO_3_	41.0	+275	−−−	−−−−	10	$TE_{011}$	[48]
LiF	9.02	+257	−−−	−−−−	7.1; 13.5	WGM	[49,53]
BaF_2_	7.35	+204	−−−	−−−−	7.9	$TE_{01\delta }$	[49]
CaF_2_	6.80	+238	−−−	−−−−	8.1; 17.5	WGM	[49,53]
SrF_2_	6.45	+230	−−−	−−−−	8.5	$TE_{01\delta }$	[49]
MgF_2_	5.48	+210	−−−	−−−−	9	$TE_{01\delta }$	[49]
NdGaO_3_	21.9	+183	−−−	−−−−	18.5	$TE_{011}$	[50]
SrTiO_4_	318	−3380	−−−	−−−−	1–10	$TE_{011}$	[50,55]
KTaO_3_	238	−3300	−−−	−−−−	1–10	$TE_{011}$	[55]
(La_2_Sr) (Al,Ta)O_3_	23.13	−0.37	−−−	−−−−	15.5	$TE_{011}$	[57]
LaAlO_3_	24.0	−−−	−−−	−−−−	10	SPDR	[52]
YVO_4_	9.36	+84	−−−	−−−−	25	$TE_{011}$	[56]

## Data Availability

Data supporting reported results can be found in references provided in the manuscript.

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
