# Peer review of "Microwave Measurements of Electromagnetic Properties of Materials"

_materials, 2021, doi:10.3390/ma14175097_

Round 1
Reviewer 1 Report
This manuscript is a complete review of material measurement using microwaves. The review is useful and well organized. Some suggestions are listed below:
1.It is suggested to add the frequency range and resonator type in Table I.
2.In the first paragraph of page 12 "... smaller samples placed at the center of the cylindrical dielectric resonator (Fig.6a), similarly as in measurements of ferroelectrics.". Fig. 6a is a typo.
Additional suggestions:
As a review article, this review is very complete. However, there are some errors. Here are my comments including the prev3. It is suggested to address the discussion about the comparison of the microwave measurement techniques about frequency, resistivity, or dielectric constant range, at the end of section 3.
4. Fig. 6 is not mentioned in the content text.
5. Some figures in the reference are almost directly used such as Fig. 11, Fig. 7, Fig. 4c. Since this is a review article, maybe the inclusion is reasonable. However, it is still suggested to confirm whether the direct usage does not conflict the rule about plagiarism.
Author Response
I would like to thank the referees for their comments and suggestions. I fully agree with Referees and I took all their suggestions into account in the revised version of the manuscript.
A detailed list of changes is as follows:
Referee 1
1.It is suggested to add the frequency range and resonator type in Table I.
I added two additional columns in Table I containing resonance type and test frequency
2.In the first paragraph of page 12 "... smaller samples placed at the center of the cylindrical dielectric resonator (Fig.6a), similarly as in measurements of ferroelectrics.". Fig. 6a is a typo.
I corrected the typo
- It is suggested to address the discussion about the comparison of the microwave measurement techniques about frequency, resistivity, or dielectric constant range, at the end of section 3.
I added the discussion about measurements of dielectrics at the end of section 4
- Fig. 6 is not mentioned in the content text.
The following sentences refer to Fig.6 in the main text: Measurements on a PbMg(1/3)Nb(2/3)O3 (PMN) ceramic sample having a diameter of 0.76 mm have been performed at a frequency of about 0.88 GHz employing modes [25]. Results of the complex permittivity measurements versus temperature are shown in Fg.6.
- Some figures in the reference are almost directly used such as Fig. 11, Fig. 7, Fig. 4c. Since this is a review article, maybe the inclusion is reasonable. However, it is still suggested to confirm whether the direct usage does not conflict with the rule about plagiarism.
Many figures in the manuscript are based on the earlier Author’s measurement data. Some of them are very similar but not identical to the figures which have been already presented in the earlier Author’s papers. I am also not sure if it does not conflict with the rule about plagiarism, although the manuscript is the review paper.
Referee 2
- Page 4 ‘Ferrites and garnets exhibit significant dispersion of the initial permeability from the acoustic to microwave frequencies.’ It is better to state the frequency range here.
I have made this correction
- Page 5 ‘In transmission/reflection methods the complex permittivity and the complex permittivity of isotropic materials.’ One of the permittivities should be permeability.
I have made this correction
- Some abbreviations used in this paper are not defined, such as ‘PWB’ and ‘YBCO’.
I have made appropriate corrections
- Page 10 ‘The higher frequency the lower becomes the term depending on the conductor losses as is shown in Fig.7.’Does the author mean ‘at’ the higher frequency?
I have changed this sentence to:
The higher frequency, the smaller becomes the effective loss tangent term depending on the conductor losses.
Referee 3
-- Section 2.3 needs references to texts of a similar level explaining in detail Debye and resonance models, Maxwell–Wagner polarization, etc. Currently, this section contains no references.
I added the famous A. von Hippel book which contains a detailed description of dielectric polarization mechanisms as a reference [4]
[4] A. von Hippel, Dielectrics and Waves, Originally published: New York: Wiley 1954
-- Fig. 7: Is frequency in MHz? Please check.
The frequency should be in Hz. I have made the appropriate corrections in Fig.7
-- Please carefully check Fig. 8, its caption, and description in the text. The panel labeling does not correspond to the discussion in the text. Panel (b) shows data for three, not two, samples. Please give more details on the difference between types of GaN samples. In the text “For sample GaN - type 1 one can observe exponential growth…”. It should be “type 2”. Is panel (a) described in the text?
I have substantially changed Fig. 8, its caption and description in the text,
-- Caption of Fig. 10 does not mention panel (b).
I have made the appropriate correction
-- The last line on the page under Fig. 10 contains a reference to Fig. 3d. It should be Fig. 4d.
I have made the appropriate correction
-- First-line below Fig. 12: “relatively large are” -> “relatively large area”
I have made the appropriate correction
With best regards
Jerzy Krupka
Reviewer 2 Report
This is a very clear and nicely organised review on different microwave measurements of electromagnetic properties of materials. The author clearly presents the setups for different measurements and their advantages/disadvantages as well as some basic concepts in microwave analysis. This review will be of wide interest to those who work with high frequency materials. The manuscript could be published as stands with some minor changes:
- Page 4 ‘Ferrites and garnets exhibit significant dispersion of the initial permeability from the acoustic to microwave frequencies.’ It is better to state the frequency range here.
- Page 5 ‘In transmission/reflection methods the complex permittivity and the complex permittivity of isotropic materials.’ One of the permittivity should be permeability.
- Some abbreviations used in this paper are not defined, such as ‘PWB’ and ‘YBCO’.
- Page 10 ‘The higher frequency the lower becomes the term depending on the conductor losses as is shown in Fig.7.’Does the author mean ‘at’ the higher frequency?
Author Response

(The authors gave the same response as above.)

Reviewer 3 Report
The manuscript is a short tutorial-level review of some successful techniques for measurements of material dielectric properties at microwave frequencies. The paper is well-written and may provide useful support for those who wish to enter the field or to understand sources of the data. In my view, the review can be published in Materials with minor revisions.
-- Section 2.3 needs references to texts of a similar level explaining in detail Debye and resonance models, Maxwell–Wagner polarization, etc. Currently, this section contains no references.
-- Fig. 7: Is frequency in MHz? Please check.
-- Please carefully check Fig. 8, its caption and description in the text. The panel labeling does not correspond to the discussion in the text. Panel (b) shows data for three, not two, samples. Please give more details on the difference between types of GaN samples. In the text “For sample GaN - type 1 one can observe exponential growth…”. It should be “type 2”. Is panel (a) described in the text?
-- Caption of Fig. 10 does not mention panel (b).
-- The last line in the page under Fig. 10 contains a reference to Fig. 3d. It should be Fig. 4d.
-- First line below Fig. 12: “relatively large are” -> “relatively large area”
Author Response

(The authors gave the same response as above.)
